# Regional mutational signature activities in cancer genomes

**Caitlin Timmons** [1,2], **Quaid Morris** [1,3]*, **Caitlin F. Harrigan** [3,4]

**1** Computational and Systems Biology Program, Sloan Kettering Institute, New York, New York, United States of America, **2** Department of Biological Sciences, Smith College, Northampton, Massachusetts, United States of America, **3** Vector Institute for Artificial Intelligence, Toronto, Canada, **4** Department of Computer Science, University of Toronto, Toronto, Canada

* morrisq@mskcc.org

**Data Availability Statement:** The GenomeTrackSig R package is available at https://github.com/morrislab/GenomeTrackSig. Data processing, analysis, and visualization code is available at https://github.com/ctimmons1/GenomeTrackSig-supplement.

## Abstract

Cancer genomes harbor a catalog of somatic mutations. The type and genomic context of these mutations depend on their causes and allow their attribution to particular mutational signatures. Previous work has shown that mutational signature activities change over the course of tumor development, but investigations of genomic region variability in mutational signatures have been limited. Here, we expand upon this work by constructing regional profiles of mutational signature activities over 2,203 whole genomes across 25 tumor types, using data aggregated by the Pan-Cancer Analysis of Whole Genomes (PCAWG) consortium. We present GenomeTrackSig as an extension to the TrackSig R package to construct regional signature profiles using optimal segmentation and the expectation-maximization (EM) algorithm. We find that 426 genomes from 20 tumor types display at least one change in mutational signature activities (changepoint), and 306 genomes contain at least one of 54 recurrent changepoints shared by seven or more genomes of the same tumor type. Five recurrent changepoint locations are shared by multiple tumor types. Within these regions, the particular signature changes are often consistent across samples of the same type and some, but not all, are characterized by signatures associated with subclonal expansion. The changepoints we found cannot strictly be explained by gene density, mutation density, or cell-of-origin chromatin state. We hypothesize that they reflect a confluence of factors including evolutionary timing of mutational processes, regional differences in somatic mutation rate, large-scale changes in chromatin state that may be tissue type-specific, and changes in chromatin accessibility during subclonal expansion. These results provide insight into the regional effects of DNA damage and repair processes, and may help us localize genomic and epigenomic changes that occur during cancer development.

## Author summary

Somatic mutations accumulate through cancer development. These mutations are the result of DNA damage and DNA repair deficiencies; in some cases these mutation sources can be recovered by attributing mutations to different mutational signatures. Mutational

**Funding:** This research was supported by an NIH/
NCI Cancer Center Support Grant P30 CA008748
to Dr Craig Thompson. Q.M. is a CCAI CIFAR chair.
C.T. was supported by the Tri-I Computational
Biology Summer Program via an NCI R25
CA233208 grant to Drs Ushma S. Neill and Michael
Berger. The funders had no role in study design,
data collection and analysis, decision to publish, or
preparation of the manuscript.

**Competing interests:** The authors have declared
that no competing interests exist.

signatures describe patterns of co-occurring substitution types (e.g., C to A) that are associated with various mutagenic processes. For example, UV radiation causes CpC to TpT substitutions, and smoking mutagens are associated with C to A substitution. Previous work has shown that mutational signature activities in cancer genomes change over the course of a cancer's development. Less is known about how their activities change across the entire genome, which is relevant for understanding the regional effects of DNA damage and repair deficiencies. We have developed a bioinformatic tool GenomeTrackSig, which uses a segmentation algorithm to estimate mutational signature activities across large chromosomal domains and identify changepoints where activities vary. We apply GenomeTrackSig to 2,203 cancer genomes and find that 426 contain changepoints. Some changepoints recur across multiple samples of the same type, and some also exhibit activity tradeoffs between signatures active in either early or late cancer development. Changepoints cannot be explained well by genomic factors that typically contribute to mutation rate variation. We hypothesize that they are driven by a confluence of factors, including large-scale changes in chromatin state that may occur over cancer evolution.

## Introduction

Cancer is a disease that develops over a lifetime through a series of somatic mutations. The vast majority of these mutations are passenger mutations, which have little effect on fitness. Multiple mutagenic processes operate in a cancer throughout its development, which in turn determine the nature of the collection of somatic mutations that accumulate. Different processes give rise to distinct mutational patterns, called *mutational signatures* [1,2]. Experimental and/or computational validation associate many mutational signatures with specific sources of DNA damage or DNA repair deficiencies [1,3]. As such, mutational signature analysis is a compelling way in which we can understand the mutational processes that shape tumor formation and development.

In nearly all tumors, multiple mutational processes contribute to generating somatic mutations. The relative contributions of these different processes can be quantified by algorithms that assign an activity (or exposure) to each detected mutational signature [4]. Recent work [5,6,7] has revealed that i) these activities vary during cancer development (i.e., tumor progression) in a cancer-type-specific way and ii) many mutational signatures are primarily active only in either early- or late-tumor progression [5,6]. We have previously described the method TrackSig [7], which uses optimal segmentation to segment an evolutionary trajectory defined by inferred cancer cell fraction (CCF). TrackSig partitions the trajectory into segments which have similar mutational signature activities by detecting changepoints. In many tumors, TrackSig uncovers changepoints in signature activity over their evolutionary trajectory, which often correspond to subclonal boundaries. We have also developed TrackSigFreq [8], an extension to TrackSig that incorporates information about mutation density at different CCF values, making the algorithm more sensitive to subclonal boundaries in the absence of signature activity changes. TrackSig and TrackSigFreq reveal changes in signature activity across evolutionary timing, but by design do not incorporate the genomic locations of point mutations. This information is relevant for mutational signature analysis, since somatic mutation rate and mutational signatures are known to be influenced by genomic and epigenomic factors that exert their effects on the scale of several nucleotides up to a megabase. These factors include sequence context, nucleosome positioning, chromatin state, and replication timing [9–18]. For example, late-replicating and heterochromatic regions accumulate more mutations than early-

replicating and euchromatic regions [15,18,19,20]. This results in somatic mutation accumulation patterns that can be attributed to chromatin state in the cell of origin. For example, in mismatch repair (MMR)-proficient tumors, the rate of CpG>TpG mutations may correlate with replication timing since MMR corrects mispairings due to 5mC deamination more efficiently in early-replicating areas [9,15].

These mutation-associated genomic features vary on scales of 1 Mb or less, so analyses of their effects on mutational signature activity are typically confined to particular loci. To our knowledge, larger scale relationships between chromosomal location and mutational signature activity have not been thoroughly investigated. Such analysis is needed, since we can expect to observe consistent changes to mutational signature activity across genomes, particularly if there exist common evolutionary events that dictate the action of mutational processes in cancers of the same type.

Wojtowicz et al. explore differential signature activity among genomic regions and introduce SigMa [21], a composite multinomial mixture model and Hidden Markov Model that assigns each mutation to a mutational signature. Transition probabilities between signatures are considered for adjacent clustered mutations, which are predominantly generated by kataegis events. However, this analysis limits the regions which can be queried for a change in mutational signature activity, excluding "sky" mutations that are more than 2 kb from the next nearest mutation. Additionally, the HMM transitions inferred by SigMa do not necessarily correspond to changes in mutational signature activity, since a transition in generating processes may occur between individual mutations in a cluster without the relative activities of those processes changing in the surrounding region. Thus, there is a dearth of methods that can discover regional changes in mutational signature activity in samples that differ with respect to mutational burden and density.

Here, we present GenomeTrackSig, which uses an optimal segmentation approach to estimate mutational signature activities across chromosomal domains and identify regions where activity changes occur. GenomeTrackSig is best-suited to samples with at least 100 mutations per chromosome, but may be run regardless of mutation density. The regional changepoints that GenomeTrackSig identifies can yield both biological and potential clinical insights. Many mutational signatures are associated with defects in DNA repair processes [1,22]; these processes can also be inhibited by clinical treatments [23]. Changepoint analysis may help us understand how these treatments' effects vary across the genome. With GenomeTrackSig, we can also explore how changes in genomic features such as copy number or chromatin state that occur during tumorigenesis affect the activity of mutational processes on a genome-wide scale. The changepoints we identify may additionally lend insight into intra-tumor heterogeneity, which presents challenges from a therapeutic perspective. Changes in signature activity may mark sections of the genome more susceptible to particular mutational processes early vs. late in tumor development. Moreover, changepoints that recur across multiple samples and tumor types can further our understanding of tumorigenesis by identifying regional DNA damages or repair deficiencies that are characteristic of a tumor type or a group of tumor types.

To examine the occurrence of mutational signature activity changes across the genome, we analyze somatic mutations of 2,203 whole genomes across 25 tumor types from the Pan-Cancer Analysis of Whole Genomes (PCAWG) consortium [23]. Using GenomeTrackSig, we identify genomic regions that recurrently exhibit mutational signature changes. These regions frequently show activity changes in signatures known to be associated with evolutionary timing. Some of these regions are shared among several tissue types, however the majority appear to be exclusive to a single tissue. By testing for associations between changepoints and copy number aberrations, kataegis events, replication timing, local chromatin accessibility, and

large-scale chromatin organization, we find that no single genomic feature alone consistently explains changepoint placement. We hypothesize that large-scale changes in chromatin state, perhaps in conjunction with evolutionary timing, may account for the observed distribution of changepoints in our samples. Our results show that regional mutational signature analysis can lend important insights into tumor development and heterogeneity.

## Results

### GenomeTrackSig robustly estimates regional mutational signature activities

GenomeTrackSig classifies single-base substitutions into one of 96 types based on substitution type and trinucleotide context. It aggregates mutations by position into bins with an equal number of mutations and represents the mutations within each bin as a mixture of multinomials, where each multinomial is a mutational signature with frequencies over all 96 mutation types. GenomeTrackSig interprets the mixture coefficients as signature activities, uses the expectation-maximization algorithm to estimate signature activities within a segment, and deploys the PELT optimal segmentation algorithm to identify "changepoints"—genomic regions where mutational signature activities change (Fig 1).

### Mutational signature activity is not constant across the genome

We construct genome-wide signature activity profiles for 2,203 tumors across 25 tumor types (Biliary-AdenoCA, Bladder-TCC, Bone-Osteosarc, Breast-AdenoCA, Cervix, CNS-GBM, Colorect-AdenoCA, Eso-AdenoCA, Head-SCC, Kidney-ChRCC, Kidney-RCC, Liver-HCC, Lung-AdenoCA, Lung-SCC, Lymph-BNHL, Lymph-CLL, Melanoma, Myeloid-MPN, Ovary-AdenoCA, Panc-AdenoCA, Panc-Endocrine, Prost-AdenoCA, Stomach-AdenoCA, Thy-AdenoCA, Uterus-AdenoCA) (Table A in S1 Text). The number and width of bins in each tumor varies from 23–2,895 and 1 Mb to 250 Mb respectively, based on the number and distribution of mutations in the tumor (S1 and S2 Figs). In total, we observe changepoints in 426 of 2,203 samples analyzed. We find no genomic changepoints in five types of tumor: Liver-HCC (N = 326), Ovary-AdenoCA (N = 113), Biliary-AdenoCA (N = 29), Panc-Endocrine (N = 76), and Myeloid-MPN (N = 31). A lack of change in signature activity could reflect an even mutational composition across the genome, insufficient number of mutations to characterize a change in signature activity, a lack of large-scale variation in chromatin accessibility, or a lack of substantial changes in mutational composition over tumor development. In the remaining 20 tumor types, at least one sample exhibits changes in signature activities across the genome,

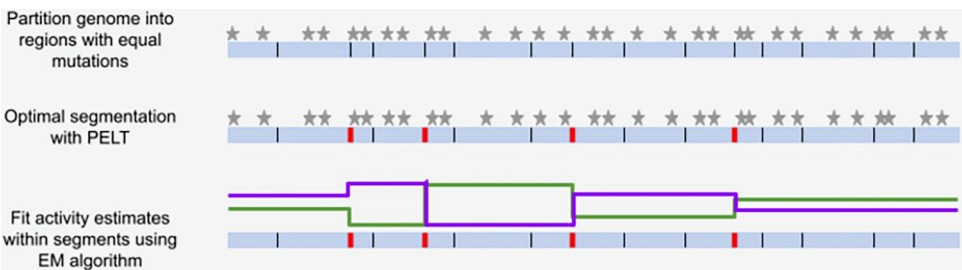

**Fig 1. Overview of GenomeTrackSig algorithm for profiling mutational signature activities across the genome.**
Stars indicate mutations (top); red bars indicate changepoints (middle); green and purple lines indicate estimated mutation signature activities; y-axis indicates exposure level (bottom). GenomeTrackSig requires at least 100 mutations per segment, fewer mutations are shown here for illustrative purposes.

indicating that large-scale shifts in signature activities depending on chromosomal location are a common but not ubiquitous phenomenon (Table 1).

## Some changepoints are shared across tissue type

We find that changepoint regions are often shared—both across multiple cancers from the same tissue and across multiple tissues. We then define a changepoint to be "recurrent" if at least 7 samples recover a changepoint in the same region. We evaluated the robustness of GenomeTrackSig's changepoint placement by resampling mutations within a genome and found that the majority of resampled profiles recover the same changepoints as the original profiles (Table D in S1 Text). In particular, all but one changepoints that recur across seven or more original profiles from the same tissue also recur across at least this many resampled profiles (Table D in S1 Text).

Of the 20 tumor types that show any changes in signature activity across their genomes, eight contain recurrent changepoint regions (Fig 2). Interestingly, the changes in signature

**Table 1. Overview of changepoints discovered across 20 cancer types.** Magnitude of each changepoint is measured as the cosine distance of the signature activity vector on either side of the changepoint. For each cancer type containing changepoints, the three signatures with the greatest absolute value of activity changes across all changepoints in that cancer are listed. Signatures are indicated as "clonal" or "subclonal" respectively based on where their activity is highest, as described by [5].

| Tumor Type | Samples with at least one changepoint / N total samples | Mean # changepoints per sample | Median # changepoints per sample | Mean changepoint magnitude | Median changepoint magnitude | SBS signatures with greatest changes | Clonal signatures | Subclonal signatures |
|---|---|---|---|---|---|---|---|---|
| Melanoma | 74/107 | 11 | 4 | 0.15 | 0.13 | 7b, 7a, 2 | 7a, 7b | 5 |
| Lung-SCC | 35/48 | 9 | 4 | 0.05 | 0.04 | 4, 13, 8 | 4 | 5, 2, 13 |
| Eso-AdenoCA | 50/97 | 5 | 4 | 0.09 | 0.07 | 17b, 40, 1 | 17 | 5, 40 |
| Lung-AdenoCA | 5/33 | 2 | 2 | 0.16 | 0.11 | 5, 13, 2 | 4 | 2, 13 |
| Colorect-AdenoCA | 21/60 | 27 | 5 | 0.04 | 0.03 | 10b, 15, 10a | 1, 44 | 18, 40 |
| Bladder-TCC | 16/23 | 6 | 3 | 0.06 | 0.05 | 13, 8, 5 | 2, 13 | 5 |
| Stomach-AdenoCA | 8/67 | 3 | 2 | 0.13 | 0.11 | 17b, 40, 15 | 1 | 18 |
| Head-SCC | 1/54 | 2 | 2 | 0.15 | 0.15 | 40, 4, 5 | 5 | 2, 13 |
| Lymph-BNHL | 29/106 | 3 | 2 | 0.2 | 0.15 | 3, 9, 6 | 9 | 17a, 17b, 40 |
| Uterus-AdenoCA | 9/51 | 41 | 48 | 0.06 | 0.05 | 15, 44, 6 | 1 | 2, 13, 40, 44 |
| CNS-GBM | 24/41 | 6 | 4 | 0.05 | 0.03 | 40, 5, 1 | 1 | 40 |
| Kidney-RCC | 8/144 | 1 | 1 | 0.08 | 0.02 | 5, 40, 13 | 40 | 1 |
| Breast-AdenoCA | 18/193 | 3 | 2.5 | 0.25 | 0.2 | 2, 5, 13 | 3, 5 | 2, 3, 13, 18 |
| Panc-AdenoCA | 12/238 | 2 | 2 | 0.38 | 0.34 | 2, 13, 1 | 1, 5 | 2, 3, 13, 17a, 17b, 18, 40 |
| Bone-Osteosarc | 13/39 | 3 | 2 | 0.31 | 0.35 | 2, 5, 3 | 2, 13 | 40 |
| Prost-AdenoCA | 11/145 | 2 | 2 | 0.17 | 0.16 | 13, 5, 40 | 5 | 40 |
| Lymph-CLL | 66/95 | 3 | 3 | 0.22 | 0.19 | 5, 40, 9 | 9 | 5 |
| Kidney-ChRCC | 5/38 | 4 | 4 | 0.24 | 0.22 | 2, 5, 40 | 5 | 1, 2, 13, 40 |
| Cervix | 13/20 | 2 | 2 | 0.08 | 0.07 | 40, 13, 5 | 1, 2, 13, 5 | 40 |
| Thy-AdenoCA | 8/29 | 2 | 1 | 0.2 | 0.15 | 5, 40, 1 | 5 | 2, 13 |

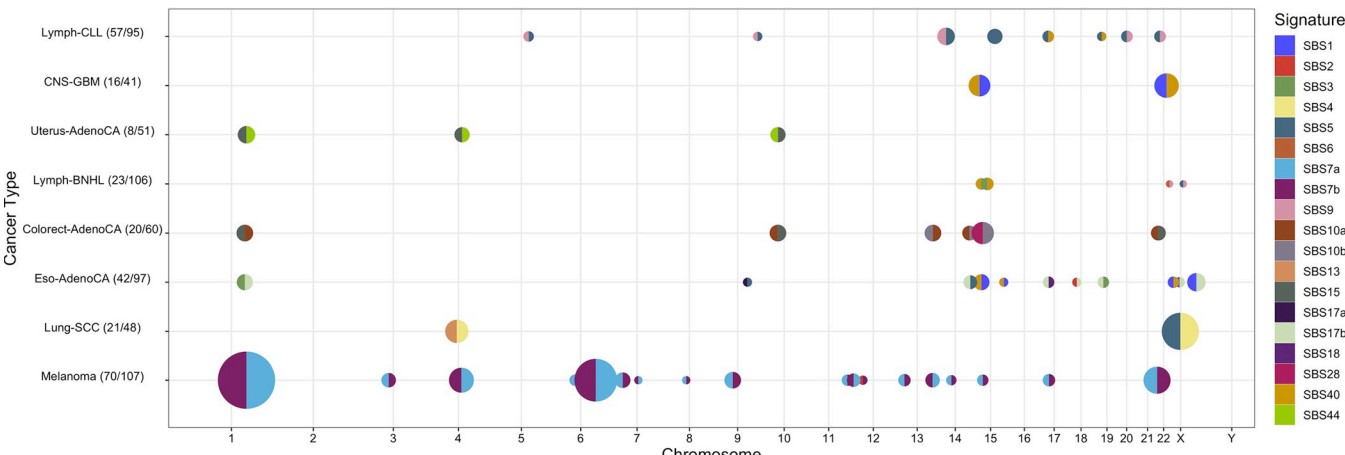

**Fig 2. Recurrent changepoint regions across eight tumor types.** 54 genomic regions include a changepoint found in seven or more samples for a given tumor type. Each of these recurrent changepoint regions are indicated by a point, where COSMIC V3 signatures [1,3] are indicated by color. Changepoints are summarized by the signature whose activity decreases the most on average (left half) and the signature whose activity increases the most on average (right half). Proportion of samples exhibiting a changepoint is encoded by size, and the number of samples affected is indicated in the tumor type label (n/N). Changepoints that appear in fewer than seven samples are omitted.

activity at these regions are often highly similar across samples of the same tissue type, both in direction and magnitude. For example, all 55 melanoma samples that have a changepoint overlapping with the chromosome 1:47,000,000–1:51,000,000 region show a decrease in SBS7b activity and increase in SBS7a activity in the direction of the numbering (Figs 1 and 2). In Uterus-AdenoCA samples, the signatures which change most at every recurrent changepoint are associated with defective DNA mismatch repair (SBS15, SBS44) (Fig 2). In other cancers, the most affected signatures are highly variable from changepoint-to-changepoint: seven different signatures are represented among the most substantial activity changes at recurrent changepoints in Eso-AdenoCA samples, including SBS1, SBS2, SBS3, SBS5, SBS17a, SBS17b, and SBS40 (Fig 2).

We also find that five recurrent changepoints are shared by two tumor types. The shared recurrent changepoint at 1:47,000,000–1:51,000,000 appears most frequently, in eight Uterus-AdenoCA samples (Fig 2) and 55 Melanoma samples (Figs 2 and 3). Another recurrent changepoint region at 1:42,000,000–1:47,000,000 is shared by 14 Eso-AdenoCA and nine Colorect-AdenoCA samples (Fig 2).

Despite the consistency of changepoint locations across multiple samples and tissues, it is unlikely that all these changepoints result from mutations in specific genes near the changepoint region. For one, changepoint regions span multiple megabases, so it is difficult to attribute the changepoint to mutations in any one gene. That said, some changepoint regions contain genes that may play a role in tumorigenesis such as Lymph-CLL changepoint regions encompassing CBL proto-oncogene B and NRP2 [24,25], and a changepoint region shared by Lung-SCC and Bladder-TCC which encompasses the DDX17 gene [26]. However, we are unable to detect any clear pattern of gene function at changepoint regions.

## Many changepoints are characterized by signatures associated with subclonal expansion

In certain cancers, we find that recurrent changepoints show a shift in activity from signatures associated with early cancer development to signatures associated with subclonal expansion. For instance, at five of eight recurrent changepoints in Lymph-CLL samples, the most

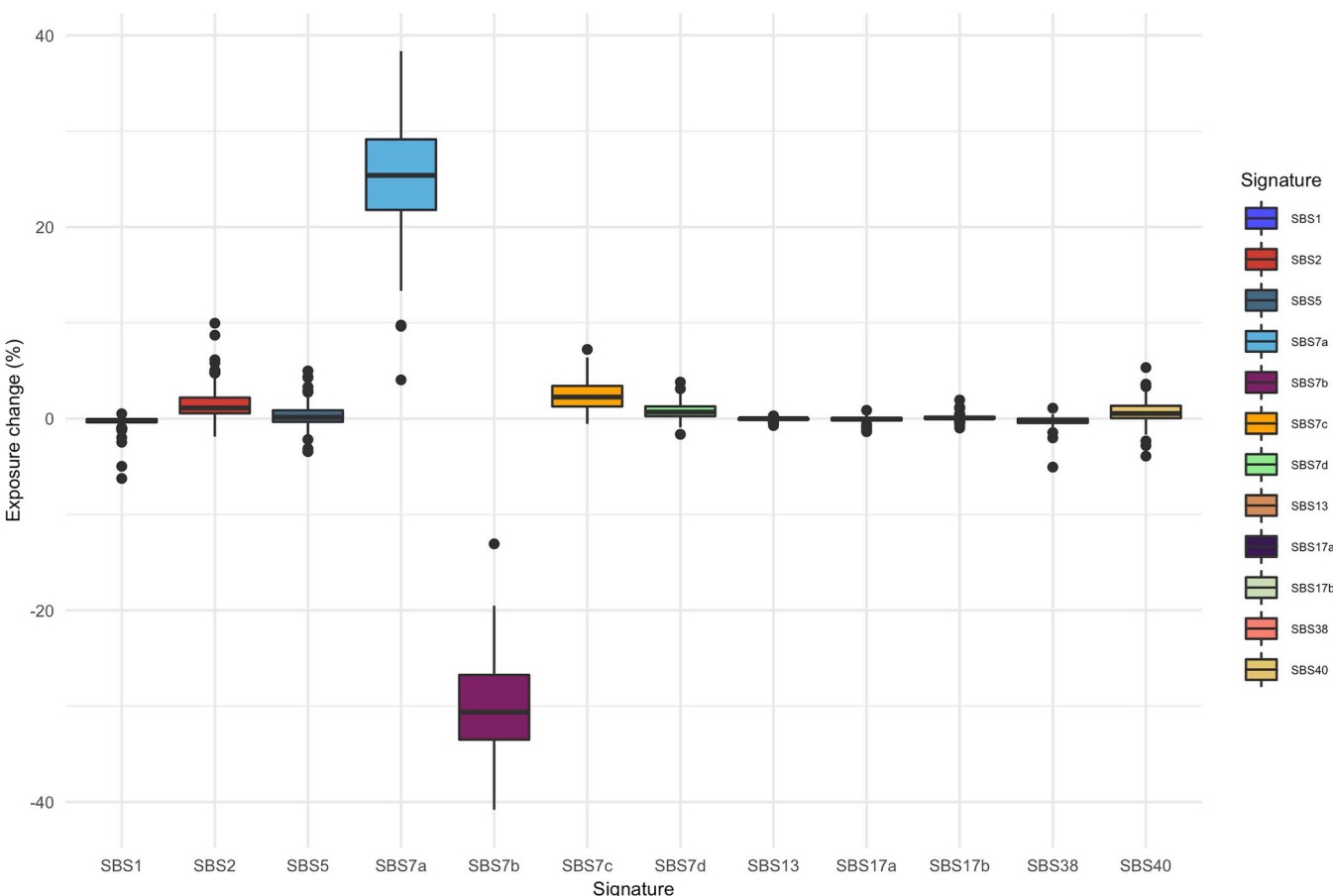

**Fig 3. Similar signature changes at the 1:47,000,000–1:51,000,000 region in 55/107 melanoma samples.** Each boxplot shows the distribution of activity changes for a particular signature at the recurrent changepoint region 1:47,000,000–1:51,000,000.

dramatic signature activity changes are increases in SBS9 and decreases in SBS5, or vice versa. We validate that these changepoints indeed reflect a shift between early and late signatures by constructing evolutionary trajectories for 90 Lymph-CLL samples using TrackSig [7]. We note that signature profiles constructed with GenomeTrackSig cannot easily be reproduced with TrackSig, because TrackSig bins mutations on cancer cell fraction rather than genomic position. When we compare position-based binning to CCF-based binning, we find that 75% of samples have no agreement between changepoints from the position-based mutation binning and changepoints from the CCF-based mutation binning. That is, the changes in signature activity with respect to genomic location that GenomeTrackSig recovers are not in general reflecting clonal identity of mutations—which may be determined using TrackSig.

Fig 4 shows the distribution of activity changes from early to late development across all samples, in which we observe a definitive decrease in SBS9, and increase in SBS5, activity. In other cancers, signatures which account for the most dramatic activity changes at recurrent changepoint regions do not show a clear timing association. For example, SBS7a and SBS7b change most dramatically across the genome in melanoma samples (Fig 3), yet do not show as strong of an association with evolutionary timing (from early-occurring to late-occurring mutations, SBS7a activity decreases by 8.7% +/- 12.2% on average and SBS7b activity increases by 1.7% +/- 9.1% on average).

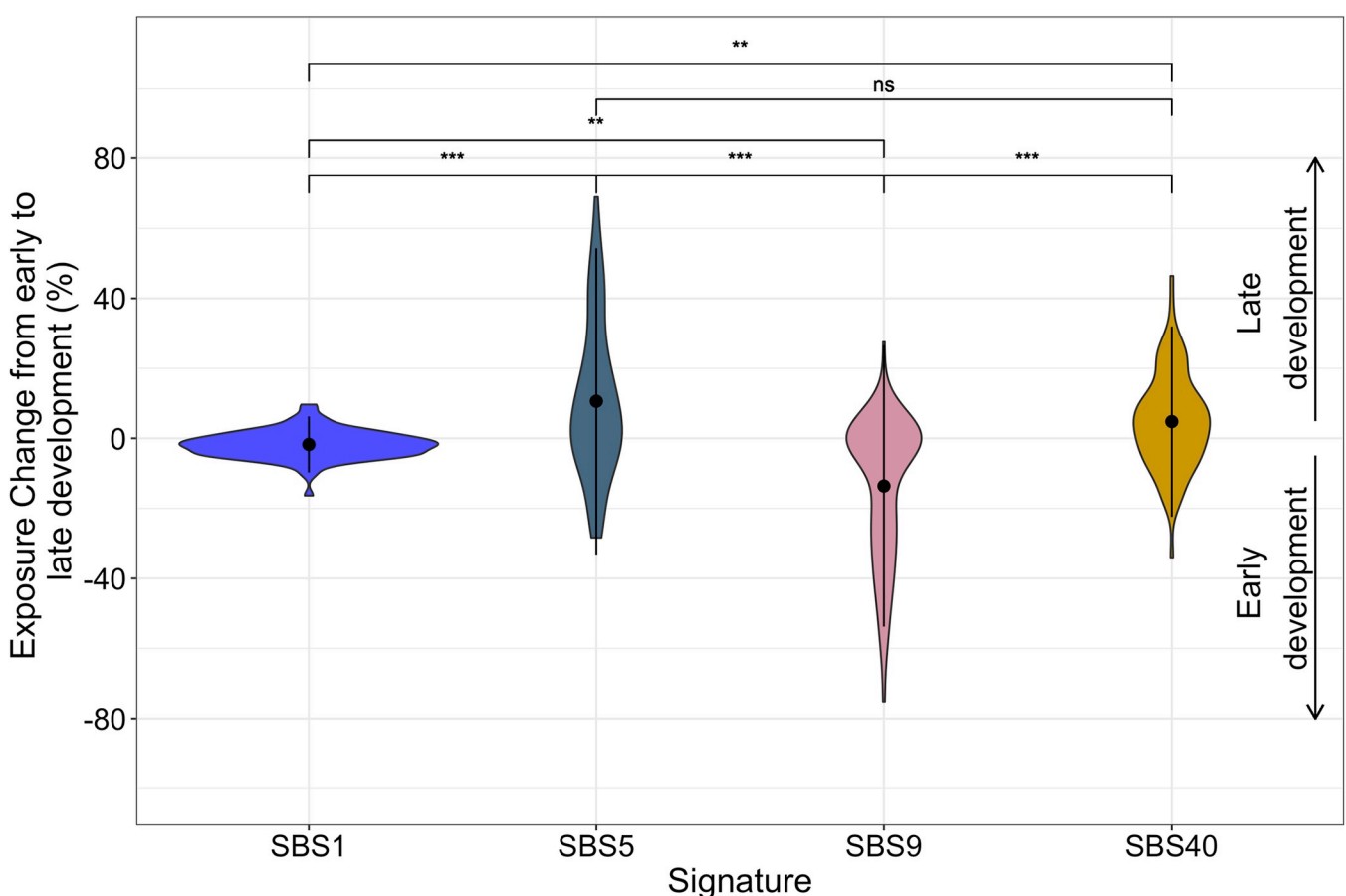

**Fig 4. Activity change in four SBS signatures during cancer evolution in 90 Lymph-CLL tumors.** Violin plots showing the distribution of signature activity changes (y-axis) from early to late stages of cancer development constructed using TrackSig [7] for 90 chronic lymphocytic leukemia samples. Dot indicates means, vertical line spans +/- one standard deviation. From early to late development, SBS1 changes on average by -1.8% +/- 4.0%, SBS5 by 10.6% +/- 21.9%, SBS9 by -13.6% +/- 20.1%, and SBS40 by 4.7% +/- 13.6%. Brackets display p-values for Kolmogorov-Smirnov tests between activity change distributions, adjusted for multiple comparisons with Bonferroni corrections. * indicates adj. p < 1e-2, ** indicates adj. p < 1e-5, *** indicates adj. p < 1e-8.

Chronic lymphocytic leukemias (Lymph-CLL) tend to exhibit high SBS9 activity early in development and show decreasing SBS9 and increasing SBS5 and SBS40 activities as subclones form (Fig 4; [5]). SBS9 activity is attributed to mutagenesis induced via DNA polymerase η. These mutations can occur in healthy lymphoid cells as part of a somatic hypermutation process, which introduces mutations to antibody-coding sequences in order to generate sequence variability and produce antibodies with higher specificity [27]. SBS9 activity is elevated in Lymph-CLL samples that possess immunoglobulin gene hypermutation [4,6], a mutational process that typically occurs early in tumor development [5,28]. Interestingly, changes in signature activity are distributed at many loci in Lymph-CLL samples, whereas in B-cell non-Hodgkin lymphoma (Lymph-BNHL), which also undergoes polymerase eta dependent somatic hypermutation, changes in signature activity are concentrated on chromosome 14, close to the immunoglobulin gene hypermutation (IGH) locus. While the etiologies of SBS5 and SBS40 are unknown, their activity correlates with patient age, and SBS5 has been associated with proliferation [1,29].

Lymph-CLL signature activity has demonstrated associations with evolutionary timing. Similar cancers also evolve in consistent patterns [5,7,23]. In keeping with this, we observe recurrent changepoints that may be indicative of important evolutionary (and shared) changes

that occur over the progression of these tumors, such as timing-dependent changes in chromatin state at these regions. Regional mutation rates and mutational signatures are influenced by chromatin state, and under normal cell growth conditions we would expect to see higher mutation rates in heterochromatic than euchromatic regions [16,17]. However, the compounding DNA damage and dysfunction in DNA repair that occurs as cancers develop could induce changes in chromatin state, making regions vulnerable to mutations late in development that were not vulnerable during early cancer formation. As such, if the chromatin state at a particular region changes during tumor evolution, this may also manifest as a regional change in signature activities.

## Changepoints are not associated with any single phenomenon

We examine the relationships between changepoint locations and changes in gene density, mutation density, tissue-specific measures of chromatin accessibility when available [30,31,32], and consensus profiles of replication timing and TAD boundaries in the human genome [33,34] to investigate what underlying genomic or epigenomic features can account for the observed distribution of changepoints. When examining the correlations between changepoints and mutation density, gene density, and chromatin accessibility, we find little evidence that any of these features alone are driving changepoint occurrences. For instance, in melanoma samples the mean correlations between changepoint occurrence and mutation density, gene density, and chromatin state as measured by DNAse-I accessibility index [32] are $r = 0.11$, $r = 0.13$, and $r = 0.12$ (Fig 5B and 5D). Changepoints occur in both gene and mutation-dense and gene and mutation-poor regions, changes in both variables are often unaccompanied by changepoints, and we can observe no obvious biased placement of changepoint with respect to these variables. Furthermore, recurrent changepoints occur across multiple samples and cancer types, where these features vary.

To further investigate if changes in mutation density affect changepoint placement, we analyze the distances between changepoints and copy number aberrations (CNAs) and the distances between changepoints and kataegis events. We conduct a randomization test to determine, for each sample, if a significant proportion of the changepoints in the sample overlap with a CNA. Overall, we find that 1/426 samples have a higher proportion of changepoints overlapping with a CNA than expected at random (Table B in S1 Text), the sole sample being a bladder cancer. We also test for potential upstream and downstream effects of CNAs on mutational signature activity. We design another randomization test to compare the distributions of genomic features, including CNAs, between the region containing a changepoint and an equivalently-sized upstream and downstream region. 8.9% of changepoints have significantly different CNA distributions between the changepoint-containing region and the surrounding regions (Table C in S1 Text). These results emphasize that the changepoints detected by GenomeTrackSig reflect genuine shifts in mutational composition, not just mutation density. Further, we find that structural variation in the genome cannot explain the observed distribution of mutational signature activity changes.

Similarly, we find that kataegis events are not a primary cause of changepoints. 2044/3059 changepoints occur within samples containing a kataegis event, defined as segments of the genome containing six or more consecutive mutations with the average distance between mutations being less than or equal to 1,000 bp. [5,36,37,38]. Changepoints and kataegis events are often distantly separated. The mean and median distances of changepoints to the nearest kataegis event are 214 Mb and 32 Mb, respectively, although some are relatively close: 357/2044 changepoints are located within 1 Mb of a kataegis event. We observe a similar pattern when analyzing recurrent changepoints. The mean and median distances to a kataegis event

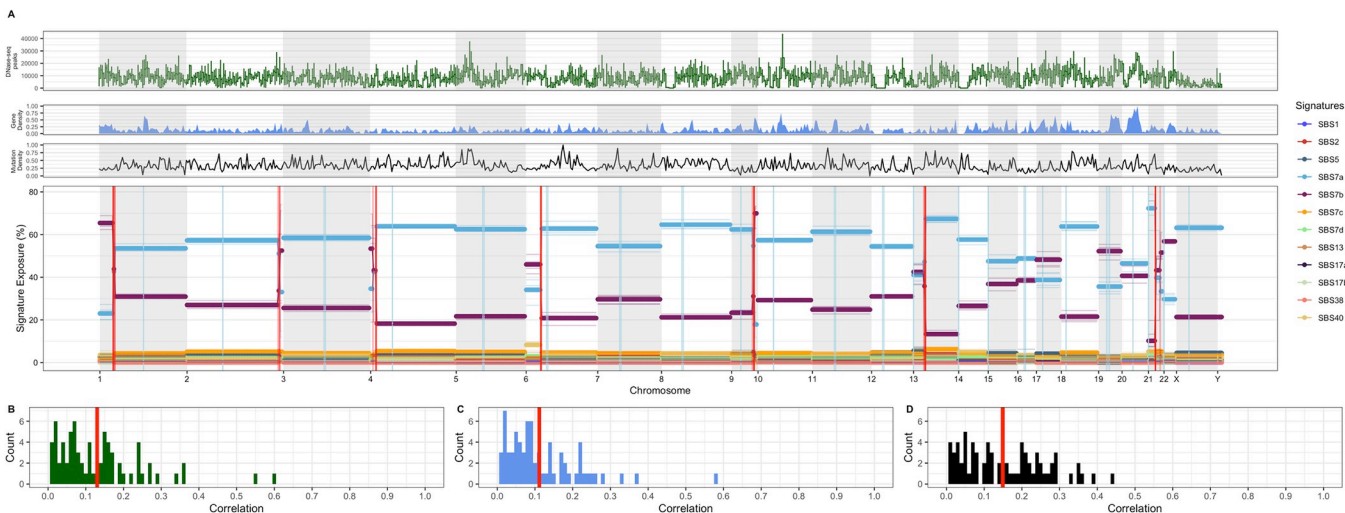

**Fig 5. The association of signature activity profiles and chromatin accessibility, gene density, and mutation density profiles. A:** Signature activity profile (chromosome-wise) for a representative melanoma sample with 168,604 mutations and 13 changepoints. Colored lines denote signature activities across bins of 200 mutations. Alternating gray and white bars denote chromosomal boundaries, and vertical blue lines show centromere positions. Red vertical lines show changepoint locations, and the opacity of these lines denotes confidence in that changepoint's location. Above, black line plot depicts mutation density at each bin across the genome. Mutation density is normalized such that the bin with the highest density throughout the genome is scaled to one and the bin with the minimum density is scaled to zero. Above, the blue area plot represents the average gene density at each bin, as determined from gene counts in hg19 [35]. Gene density is normalized in the same manner as mutation density. Above, the green line plot depicts chromatin accessibility throughout a primary melanocyte genome as determined via DNAse-I accessibility index [33]. Horizontal lines at each bin show the average chromatin accessibility across each bin, and vertical lines depict the range of chromatin accessibility values within each bin. **B, C,** and **D** show the distribution of correlation values between changepoint locations and DNAse-I accessibility index, gene density, and mutation density, respectively, across the 74 melanoma samples which contain changepoints. Mean correlation is highlighted in red on each plot.

among recurrent changepoints are 201 Mb and 18 Mb respectively, and 173/783 changepoints are located >1 Mb away from their nearest kataegis event. With the same general randomization test used in our CNA analysis, we find that in 7/426 samples, a significant proportion of changepoints overlap with a kataegis event (Table B in S1 Text). Furthermore, 5.9% of changepoints show significantly different distributions of kataegis events between the changepoint-containing region and its surrounding regions (Table C in S1 Text). These results are consistent with kataegis being one cause of a changepoint but not their primary cause. One reason for the partial association of kataegis and changepoints may be that the vast majority of kataegis events involve APOBEC deaminases, which are represented by SBS2 and SBS13 [1,37]. In this case, the hypermutation would result in a significant change in the local activity of these signatures, which would be detected by GenomeTrackSig. The localized hypermutation of the IGH locus in Lymph-BNHL may result in a SBS9-enriched kataegic event, which may explain both the high incidence of kataegis in Lymph-BNHL as well as the proximity of kataegis to the recurrent changepoints in that cancer (mean and median distances = 7 Mb and 6 Mb, respectively).

Lymph-BNHL exhibits at least one polymerase eta-driven hypermutation hotspot on every chromosome [23], however changepoints are almost exclusively found on chromosome 14 (Fig 2). In contrast, Lymph-CLL contains eight recurrent changepoints across different chromosomes (Fig 2) but relatively few kataegis foci [23]. Although these two cancers have commonalities in their activities of SBS1, SBS9, SBS5, and SBS40 [1] as well as being the only two cancer types known to display polymerase η-driven kataegis [23], within them we observe nearly opposite relationships between GenomeTrackSig changepoints and kataegis foci. In summary, kataegis has some association with GenomeTrackSig changepoints but it does not

explain the majority of changepoints, and its presence does not always give rise to a changepoint.

To investigate how other genomic and epigenomic features might contribute to our observed distribution of changepoints, we test for effects of replication timing, chromatin accessibility, and 3D chromatin organization on changepoint placement. These features could drive changepoint placement because under normal cell growth conditions, late-replicating and heterochromatic regions tend to accumulate more mutations than early-replicating and euchromatic regions [13,16,17]. We obtain a genome-wide consensus replication timing profile, [33] and find that no changepoints exhibit significantly different replication timing profiles between the changepoint-containing segment and the upstream/downstream segments (Table C in S1 Text). We can thus conclude that replication timing is not likely a significant driver of changepoints in the samples tested.

Furthermore, we examine whether changepoint-containing regions and their surrounding regions show significant differences in chromatin accessibility. When available, we compare distributions of tumor type-specific ATAC-Seq peaks [30] and tumor type-specific A/B compartments in chromatin structure as determined by Hi-C assays [31]. We find that 11.9% of changepoints across the 15 applicable tumor types have significantly different chromatin accessibility (as measured by ATAC-Seq peaks) between the changepoint-containing region and surrounding regions, and 24.2% of changepoints across the eight applicable tumor types have significantly different A/B compartment structure between the changepoint-containing region and surrounding regions (Table C in S1 Text). These findings suggest that regional changes in chromatin state contribute to some, but not all, changepoints. Chromatin state appears to have a greater influence on regional mutational signature activity changes than CNAs or kataegis events, consistent with our hypothesis that increasing dysfunction in DNA repair and epigenetic organization over the course of cancer development triggers changes in chromatin state that lead to regional shifts in mutational composition and signature activity.

Since changepoints typically demarcate chromosomal domains greater than 1 Mb, we also test for associations between changepoints and topologically associated domains (TADs), which on average span about 1.15 Mb in humans [39,40,41,42]. Using the same general analysis performed for CNAs and kataegis events, we identify samples with a significantly higher proportion of changepoints overlapping with one or more of the consensus TAD boundaries estimated for the human genome [33] than would be expected at random. In total, 61/426 samples show a significant degree of changepoint/TAD overlap (Table B in S1 Text). Genomic rearrangements in cancer cells can disrupt TAD boundaries or fuse discrete TADs [43,44], so the true distribution of TAD boundaries in our samples may diverge from the consensus estimate. Still, our findings are consistent with the hypothesis that large-scale chromatin organization plays a role in driving mutational signature activity changes across the genome.

## Discussion

Here we introduce a new method, GenomeTrackSig, to detect region-specific changes in mutational signature activity within cancer genomes. Using this method, we demonstrate frequent changes in mutational signature activities over large chromosomal domains in a variety of cancers. We also find a surprising number of recurrent changepoints in signature activities shared across cancers of the same type and among cancers of different types.

The scale, >1 Mb, at which GenomeTrackSig can generally detect changes in mutational signature activity is larger than many regional factors already known to influence mutation rates, which makes it interesting that we see a high number of changepoints. For example, regional mutation rate varies due to changes in replication timing, differential activity of DNA

repair mechanisms, and chromatin accessibility [12–19]. In samples with many mutations, in which a changepoint can demarcate smaller regions (~ 1 Mb in size), we may be able to attribute changes in signature activities to some of these factors that cause regional differences in the somatic mutation rate, especially if these factors change substantially over tumor development (see below). However, our effort to quantify this phenomenon using readily-measured genomic features–gene density, mutation density, copy number variants, kataegis, replication timing, and chromatin accessibility (Fig 5 and Tables B and C in S1 Text)–establishes that the signature changes we detect reflect a genuine shift in underlying mutational distributions and cannot be fully explained by these local, smaller-scale factors alone (Fig 5).

Perhaps, instead our changepoints demarcate large-scale chromosomal organization, with each large domain having its own underlying mutational distribution driven by domain-specific DNA damage and repair dynamics. Given that signature changes at recurrent changepoints are often consistent within a tissue yet variable across tissues, it is possible that such chromosomal domains can occur at similar locations in multiple tissue types yet exert tissue-specific effects on the mutational landscape. One intriguing possibility is that these large-scale domains are delineated by the 3D organization of the cancer genome–representing, for example, larger topologically associating domains (TADs) [40–43]. Our analysis of associations between changepoints and TAD boundaries reveals that some changepoints may be driven by shifts in gene regulation and chromatin organization across 3D domains, although sample-specific TAD disruptions and their effects on changepoint placement remain an open question.

The signature dynamics at the most common changepoint in melanoma samples support our hypothesis that changepoints are driven by chromatin state variation over large scales. The changepoint region at 1:47,000,000–51,000,000 is found in 77 samples from two tumor types, 55 of those melanomas. As shown in Fig 3, all 55 melanomas display strikingly consistent decreases in SBS7b activity and increases in SBS7a activity at this changepoint. An analysis of the genomic properties influencing mutational signature activities by [16] using a *de novo* signature extraction method, identifies two signatures exclusively occurring in skin cancers which highly resemble SBS7a and SBS7b in terms of their mutational distributions and associations with UV exposure. This study finds that the SBS7a-like signature has high activity in quiescent chromatin, whereas SBS7b-like is enriched in active chromatin and has a strong transcriptional strand bias. This differential signature activity across chromatin states may reflect different operative DNA repair mechanisms. In this scenario, SBS7a-like activity reflects UV damage cleared by global genome nucleotide excision repair (GG-NER), which operates in quiescent and active regions, and SBS7b-like activity reflects damage cleared by a combination of GG-NER and transcription-coupled nucleotide excision repair (TC-NER), the latter of which operates in open chromatin and is activated by template strand lesions on actively transcribed genes [16]. Therefore, the activity shifts between SBS7a and SBS7b that we observe at recurrent changepoint regions in melanomas may reflect large-scale changes in chromatin state, coupled with changes in active DNA repair processes.

Based on signature activity profiles from Lymph-CLL samples, we also hypothesize that large-scale changes in chromatin state sometimes occur in a timing-dependent manner. We observe changepoints in which early-development signatures decrease in activity and late-development signatures increase in activity, and vice versa. Cancer development is marked by a redistribution of the mutational landscape, as exposures to mutagens and DNA repair failures strip active genes in euchromatic regions of the preferential repair that they normally receive [15]. Thus, we may see higher activity of late-development signatures in regions that normally have a low mutation rate, but which become vulnerable to mutation during subclonal expansion when different mutational processes are active than were early on. Conversely, late-

replicating regions of closed chromatin might display higher activity of early-development signatures since these tend to have higher mutation rates under normal conditions.

Via mutational signature analysis, we identify that mutational signature activities change over chromosomal domains. These changes can be highly consistent on multiple levels. We hypothesize that, among other factors, our method is detecting the impacts of large-scale regional changes in chromatin state. The signature dynamics at recurrent changepoints in melanoma samples provide compelling evidence that regions on either side of a changepoint differ in terms of their chromatin accessibility and DNA repair dynamics (Fig 3; [16]). Furthermore, changepoints that appear in multiple tissue types raise the possibility that such wide scale epigenomic changes are common events during cancer development. These results call for further exploration into how factors like chromatin state and DNA repair mechanisms vary over wider domains, and which elements of these changes are characteristic of many cancers and which are tissue-specific. Interestingly, changepoints sometimes reflect a shift in activity between signatures characteristic of early tumor development and signatures characteristic of subclonal expansion. Thus, we hypothesize that large-scale changes in chromatin accessibility may also occur in a timing-dependent manner and can demarcate regions that are more likely to be impacted by mutations early or late in a cancer's evolutionary trajectory. This finding may be clinically relevant given that intra-tumor heterogeneity is a mechanism of therapeutic resistance and therefore presents significant challenges for treatment [45,46,47,48]. Therefore, genome-wise mutational signature analysis can help us further characterize and localize the genomic and epigenomic changes that occur during tumor development.

## Methods

### Ordering the mutations

Mutations are ordered by chromosomal coordinates. GenomeTrackSig may be run either i) chromosome-wise or ii) genome-wise. In the chromosome-wise setting, each chromosome is segmented independently, and mutation ordering is strictly respective to its coordinate on the chromosome of its locus. In the genome-wise setting, chromosomes are treated as if laid end-to-end in order [1–22, X, Y]. Mutations are ordered according to both chromosome number, and coordinate. Because this ordering is not reflective of biological proximity, but rather a numbering convention, we do not recommend using the genome-wise setting unless the total number of mutations in a sample is small (<100 per chromosome). In the genome-wise setting, bootstrapping may be performed by randomizing the order of chromosomes. The genome-wide approach can capture signature activities that span multiple chromosomes thereby potentially reducing noise in activity estimates at chromosomal ends. On the other hand, the chromosome-wise approach may be more sensitive to activity changes within a chromosome and substantially reduces runtime on samples with many mutations because each chromosome can be segmented in parallel (S3 Fig).

### Choice of active signatures

The set of signatures to fit in each sample was selected based on those reported by Alexandrov et al. [1] to be active above 5% in any PCAWG sample of the corresponding tissue-type classification.

### Choice of bin size

By default, we partition the set of mutations in a genome into bins containing 100 mutations each, where the size of the chromosome region (i.e., number of basepairs) spanned by a bin

will vary depending on mutation density. We require each segment in the optimal segmentation solution to include at least one bin. The last bin in the segmentation (at the end of each chromosome for the chromosome-wise setting, and at the end of the last chromosome for the genome-wise setting) is allowed to contain more than 100 mutations, when the total number of mutations cannot be perfectly binned.

To improve execution time, we used a bin size of 200 mutations for cancers with the highest mutational burdens, namely Melanoma, Lung-SCC, Eso-AdenoCA, Lung-AdenoCA, and Colorect-AdenoCA, and the default bin size of 100 mutations for all other cancers. We compared changepoint placement across varying bin sizes (100, 150, 200, 300) to establish that changepoint location is relatively insensitive to bin size choices above the recommended minimum of 100 mutations (S4 Fig).

## Detecting changepoints using optimal segmentation

GenomeTrackSig, like its predecessor TrackSig [7], identifies changes in mutation signature activity across the genome using the Pruned Exact Linear Time (PELT) optimal segmentation algorithm. The key difference between GenomeTrackSig and TrackSig is the axis along which mutations are ordered, but the changepoint detection algorithm remains unchanged. A formal description of changepoint detection by the TrackSig algorithm is provided in S1 Note. TrackSig estimates a cancer's evolutionary trajectory by detecting changes in mutational signature activity across pseudo-time by ordering mutations via estimated cancer cell fraction. We adapt the TrackSig framework by binning mutations by proximity instead of CCF, such that instead of considering evolutionary timing of mutations, we can detect changes in signature activity across chromosomal coordinates. An illustration of applying GenomeTrackSig to segment a genome is shown in Fig 1. Although GenomeTrackSig uses the same changepoint detection methods, it extends the capabilities of the original TrackSig algorithm to explore how mutational signature activities vary across chromosomal regions in a variety of tumors because of the modified binning approach.

In accordance with the original TrackSig algorithm, we represent a sample containing N mutations as an N x K matrix where each mutation is given as a one-hot-encoded binary vector over K = 96 mutation types. This matrix is in turn represented as a mixture of multinomials, in which the mixture coefficients are interpreted as the signature activities.

We use the EM algorithm [48] to iteratively update the estimates of (E-step) the responsibilities of mutations assigned to signatures and (M-step) the mixture coefficients of signatures in each segment.

Changepoints are identified using the Pruned Exact Linear Time (PELT) segmentation algorithm [49]. PELT combines dynamic programming and branch and bound search to perform optimal segmentation which is efficient in the number of mutations. At each iteration, PELT considers adding a new changepoint out of the set of available regions and scores the changepoint by refitting the activities in each region. Over-segmentation is penalized using the Bayesian Information Criterion (BIC). Previous work by [1] characterized canonical signatures active within the PCAWG data.

## Bootstrapping changepoint location

We evaluate confidence in signature activity estimates and changepoint placements for an individual genome using bootstrapping. We sample with replacement the mutations within each bin. This method is applicable for profiles constructed with either the genome-wise or chromosome-wise approaches described above. We construct signature profiles for all the PCAWG tumors with five bootstrap samples using this method.

### Determining recurrent changepoints

To determine which changepoints within samples from the same tumor type overlap, we fit a kernel density estimate across the vector of genomic locations covered by each changepoint and constructed a possible range for that changepoint location, which contains the changepoint locus identified by GenomeTrackSig, +/- one standard deviation of the density function. We then overlay all changepoint ranges from that tumor type and count how many changepoints fall within a sliding window across the genome. We then define a 'recurrent changepoint region' for each tumor type as the center of the region where at least seven samples share an overlapping changepoint; i.e. the window within a region in which the maximum number of changepoints fall. This threshold corresponds to an elbow across several tissue types (S5 Fig).

### Activity of early- and late-clonal signatures in Lymph-CLL

We use the TrackSig R package (v0.2.0) to identify early- and late-clonal signatures in Lymph-CLL samples. The set of active signatures to fit activities for are selected according to Alexandrov et al. [1]. These are: SBS1, SBS5, SBS9, SBS40.

### Comparing position-based binning to CCF-based binning

We use the TrackSig R package (v0.2.0) to identify subclones using CCF-based segmentation. Active signatures are identified using the detectActiveSignatures() method and changepoint detection is run using default parameter settings. We compute the number of subclones represented in each bin of mutations aggregated by position with GenomeTrackSig. We compare the locations of bins where GenomeTrackSig finds changepoints with the locations of bins where the subclone that accounts for the majority of mutations changes.

### Assessing changepoint robustness through resampling

We evaluate the robustness of GenomeTrackSig's changepoint placement by resampling mutations within a genome and comparing the changepoints that GenomeTrackSig identifies in the resampled profile to those in the original profile. We restrict our analysis to samples with recurrent changepoints (Fig 2) since these changepoints are most relevant for understanding tissue-specific mutational patterns. For each sample, we keep bin boundaries constant and sample with replacement from the mutations within each bin. We construct 20 resampled signature activity profiles per genome using the same bin size as the original profile. We then compare, for a given cancer type, how many of the recurrent changepoints in original profiles are also recurrent (found in at least seven samples) across the resampled profiles. Additionally, by comparing each pair of original and resampled profiles, we calculate the number of resampled profiles that recover the same recurrent changepoints as their original counterpart. These two measures give us a tissue-level and sample-level view, respectively, of GenomeTrackSig's reliability in identifying recurrent changepoints. Furthermore, we calculate the proportion of changepoints that are found in more than one bootstrap sample within both the original and resampled profiles. We refer to these as "high-confidence changepoints" (Table D in S1 Text).

### Testing for effects on changepoint placement

We use two resampling methods to test for potential effects that genomic features other than mutation counts have on changepoint placement. The first method tests the significance of overlap between changepoints and genomic features, which determines if changepoints frequently co-occur with particular features. We use this test for three sparse/discrete genomic features: copy number aberrations (CNAs), kataegis events, and TAD boundaries. The second method tests

whether the distribution of a genomic feature is significantly different between a changepoint-containing region and similar, but changepoint-free regions upstream and downstream of the changepoint. This test determines if changepoints are driven by properties of nearby genomic regions. We use this test for two discrete genomic features—CNAs and kataegis events—and four continuous genomic features: chromatin accessibility as measured by ATAC-Seq peaks, chromatin accessibility as measured by A/B compartments, gene density, and replication timing.

Each test uses the same general method across different genomic features. Below we describe the general method used to test for significant overlap between changepoints and discrete features. We begin with a genome-wide signature activity profile with a set of changepoints. We compare the chromosomal regions spanned by the changepoints and the chromosomal regions spanned by the genomic feature. If any part of a changepoint region overlaps with any part of a feature region, we say that the changepoint overlaps with the feature. We calculate the proportion $p$ of changepoints in the sample that overlap with the genomic feature. We then construct a null distribution of 10,000 observations of random changepoint sets and compare $p$ to the proportion of random changepoints which overlap with the genomic feature. We generate random changepoint sets by sampling an equal number of changepoints to the original set, where each changepoint has a random location drawn from the bins in the original signature profile. Overlaps between random changepoints and the genomic feature are determined in the same manner as used for the sample changepoints. Using a significance cutoff of $\alpha = 0.05$, we determine that if $p$ is significantly greater than we would expect at random, then the discrete genomic feature is a significant driver of changepoints in the sample. We repeat this process for all applicable samples.

Here, we describe the general method used to test whether a genomic feature has a significantly different distribution between a changepoint-containing region and an equal-size upstream or downstream region.

We begin with a genome-wide signature activity profile with a set of changepoints. For each changepoint region $c$, we identify the closest downstream changepoint region which we refer to as $c_d$. If there is no closest downstream changepoint (ie. c is the first changepoint on the chromosome) then we take the chromosomal boundary as $c_d$.

We then find the midpoint between the two changepoints, which we call $m_d$. We then define two segments, which we call the changepoint-containing segment and the downstream segment. The changepoint-containing segment is centered at the middle of c, and its width is equal to the distance between c and $m_d$. The downstream segment is centered at $c_d$ and has the same width as the changepoint-containing segment. We find the distributions of the genomic feature within each segment defined above. We compare their distributions using either the Kolmogorov-Smirnov test [50] for continuous measurements or Fisher's exact test [51] for counts. This test yields a p-value, $p_{cd}$.

We then construct a randomization distribution with 10,000 observations to test if $p_{cd}$ is significantly smaller than we would expect at random. To build this distribution, we generate 10,000 random changepoint sets. We generate random changepoint sets by sampling an equal number of changepoints to the original set, where each changepoint has a random location drawn from the set of bins in the original signature profile. Within each random changepoint set, we draw for each changepoint a random downstream changepoint. The downstream changepoints are separated from the changepoint by at least one bin since GenomeTrackSig does not allow for changepoints at adjacent bins. We define the changepoint-containing segment and downstream segment for the random changepoint and its downstream counterpart, and test for differences in distributions, in the same manner for the random sample as for the original sample. These tests yield a distribution of 10,000 p-values. Using a significance cutoff of $\alpha = 0.05$, we test if $p_{cd}$ is significantly smaller than we would expect by chance.

We conduct the same analysis to test for differences in distributions between the changepoint-containing region and an equivalent upstream region. In this case, we identify the nearest upstream changepoint, or chromosomal boundary if this is the last changepoint on the chromosome, and define the changepoint-containing region and upstream region using these locations. This randomization test yields another p-value, $p_{cu}$. For each changepoint, if either one of $p_{cu}$ or $p_{cd}$ are significant, we report that the changepoint has significantly different feature dynamics between the changepoint-containing region and the surrounding regions (Table C in S1 Text).

## Determining effects of copy number aberrations on changepoint placement

We obtained consensus copy number calls for each sample from the PCAWG data portal [23]. We define copy number aberrations (CNAs) as genomic regions with ploidy levels other than diploid. Each dataset only contains copy number calls for autosomes, so we filter signature activity profiles and changepoint sets to remove information on the X and Y chromosomes.

We determine which samples have a significant proportion of changepoints overlapping with CNAs using the first randomization method described above for overlap between changepoints and discrete genomic features (Table B in S1 Text). We also use the second randomization method described above to test for significantly different CNA distributions between changepoint-containing regions and surrounding regions (Table C in S1 Text). We generate CNA distributions by summarizing the number of CNAs within each 1 Mb sliding window across the genome. We use Fisher's exact test [51] to test for differences in distributions.

## Determining effects of kataegis events on changepoint placement

Kataegis events are regions of the genome containing six or more consecutive mutations with the average distance between mutations being less than or equal to 1,000 base pairs [6,36,37,38]. For each sample, we identified kataegis events using the rainfallPlot function within the Maftools R package [36]. For our analyses, we consider only samples which contain both changepoints and kataegis events.

We determine which samples have a significant proportion of changepoints overlapping with kataegis events using the first randomization method described above for overlap between changepoints and discrete genomic features (Table B in S1 Text). We also use the second randomization method described above to test for significantly different kataegis event distributions between changepoint-containing regions and surrounding regions (Table C in S1 Text). We generate kataegis event distributions by summarizing the number of kataegis events within each 1 Mb sliding window across the genome. We use Fisher's exact test [51] to test for differences in distributions.

## Determining effects of TAD boundaries on changepoint placement

We obtain a consensus estimate of topologically-associated domains (TADs) across the human genome from [34]. TAD boundaries are the genomic regions in between two distinct TADs. We determine which samples have a significant proportion of changepoints overlapping with TAD boundaries using the first randomization method described above for overlap between changepoints and discrete genomic features (Table B in S1 Text).

## Determining effects of chromatin accessibility on changepoint placement

We obtained cancer-type-specific consensus ATAC-Seq data from [30]. Cancer-type-specific information is available for Bladder, Breast-AdenoCA, Cervix, CNS-GBM, Eso-AdenoCA,

Head-SCC, Kidney-RCC, Kidney-ChRCC, Lung-AdenoCA, Lung-SCC, Melanoma, Prost-AdenoCA, Stomach, Thy-AdenoCA, and Uterus-AdenoCA samples. Each dataset contains numerical ATAC-Seq peak calls across contiguous genomic regions. We use the second randomization method described above to test for significantly different ATAC-Seq peak distributions between changepoint-containing regions and surrounding regions (Table C in S1 Text). We use a Kolmogorov-Smirnov test [50] to test for differences in distributions.

We also obtain datasets with cancer-type-specific chromatin A/B compartment information from [31]. Cancer-type-specific A/B compartment measures are available for Bladder, Breast-AdenoCA, Colorect-AdenoCA, Lung-AdenoCA, Lung-SCC, Prost-AdenoCA, Thy-AdenoCA, and Uterus-AdenoCA samples. Each dataset contains contiguous 1 Mb genomic regions denoted either as A compartment (associated with open chromatin) or B compartment (associated with closed chromatin). We recode chromatin compartment to a numerical variable and use the second randomization method described above to test for significantly different A/B compartment distributions between changepoint-containing regions and surrounding regions (Table C in S1 Text). We use Fisher's exact test [51] to test for differences in distributions, since each distribution contains counts of A compartment regions and B compartment regions.

## Determining effects of gene density on changepoint placement

We calculate gene density at 1 kb sliding windows across the genome using the tileGenome and countOverlaps functions from the GenomicRanges R package [12], using the genome annotation for human reference genome hg19 from [35]. We use the second randomization method described above to test for significantly different gene density distributions between changepoint-containing regions and surrounding regions (Table C in S1 Text). We use a Kolmogorov-Smirnov test [50] to test for differences in distributions.

## Determining effects of replication timing on changepoint placement

We obtain replication time profiles across contiguous genomic regions for 326 induced pluripotent stem cell lines [33] and take the median replication time across cell lines at each segment. Note that profiles are highly reproducible across samples (median r = 0.93). We use the second randomization method described above to test for significantly different median replication timing distributions between changepoint-containing regions and surrounding regions (Table C in S1 Text). We use a Kolmogorov-Smirnov test [50] to test for differences in distributions.

## Supporting information

**S1 Text. Table A in S1 Text.** Abbreviations of tumor types analyzed in the study. **Table B in S1 Text: Co-occurrence of changes in mutational signature activities and copy number aberrations, kataegis events, and TAD boundaries across 20 tumor types.** For each tumor type, we conducted a randomization test to determine how many samples have a significant proportion of their changepoints overlapping with one of these features. Randomization tests were conducted by generating 10,000 random samples with the same number of changepoints as the original samples with changepoint locations randomized. Each random set of changepoints was compared to the original sample copy number profile, kataegis profile, or consensus TAD map to calculate the proportion of changepoints overlapping with a given feature. This set of proportions formed the null distribution against which the sample proportion was compared to determine a p-value ($\alpha = 0.05$). **Table C in S1 Text: Distributions of genomic/epigenomic features compared between regions containing changepoints and upstream**

**and downstream regions across 20 tumor types.** Each column shows the proportion of changepoints in a cancer with significantly different genomic feature distributions between the region containing the changepoint and the upstream and/or downstream region. For each changepoint, feature distributions were compared between two equally-sized adjacent segments, one centered in the middle of the changepoint region and the other centered in the middle of the span between the changepoint and the nearest upstream/downstream changepoint or chromosomal boundary, if nearer. Distributions were compared using Fisher's exact test for CNA, kataegis, and A/B compartment measurements, and using the Kolmogorov-Smirnov test for ATAC-Seq, DNAse-Seq, gene density, and replication timing measurements. For each sample, 10,000 randomly-located changepoints were generated and the same distributions were compared to form a null distribution of p-values. The proportion of null distribution p-values less than or equal to the sample changepoint was recorded as the final significance measure ($\alpha = 0.05$). Cancer-specific DNAse-Seq measurements obtained from Polak et al. 2015 were available only for primary melanocytes and are thus not shown as a column in the table. We found that 77 / 786 melanoma changepoints have significantly different chromatin accessibility—as measured by DNAse-I accessibility index—between the region containing the changepoint and the upstream and/or downstream regions. **Table D** in S1 **Text**: **GenomeTrackSig changepoint recovery on original profiles compared to resampled profiles.** Analysis restricted to samples which display recurrent changepoints. As described in Methods, we run GenomeTrackSig with five bootstrap samples to create 'original' signature profiles. We run GenomeTrackSig with 20 bootstrap samples to create 'resampled' signature profiles. Matching resampled profiles to original profiles, we calculate the number of high-confidence changepoints (changepoints found in more than 1 bootstrap sample) recovered and the number of recurrent changepoints recovered. Within a cancer type, we compare all resampled profiles to all original profiles to calculate the overall number of recurrent changepoints in original profiles recovered by at least 7 resampled profiles.
(DOCX)

**S1 Note. Formal description of changepoint detection.**
(PDF)

**S1 Fig. Number of bins per sample across cancer types.** Stacked bar chart depicting the distribution of bin numbers across samples, colored by cancer type. Minimum number of bins is 23 (Melanoma, Lung-AdenoCA, Lymph-BNHL, Kidney-RCC, Prost-AdenoCA, Lymph-CLL, Kidney-ChRCC, Thy-AdenoCA) and maximum number of bins is 2895 (Colorect-AdenoCA). Sample size and geometric mean TMB is shown for each cancer type.
(TIFF)

**S2 Fig. Distribution of bin widths by cancer type.** Boxplots showing the range of bin widths, in megabases, for all samples analyzed in the study. Boxplots are outlined according to the type of bin plotted, either all bins or bins containing changepoints. Sample size and geometric mean TMB is shown for each cancer type. Boxplots are colored according to geometric mean TMB.
(TIFF)

**S3 Fig. Comparison of genome-wise (A) and chromosome-wise (B) activity profiles constructed by GenomeTrackSig.** Input data is a Lung-SCC genome with 78,839 mutations. A bin size of 200 mutations was used and 5 bootstraps were performed for each experiment. Each point is a signature activity estimate at one bin of mutations. Alternating gray and white bars distinguish chromosomes and blue vertical lines show centromere positions. Red vertical

lines denote changepoints, and the opacity of changepoints represents their bootstrap support. (TIFF)

**S4 Fig. Lung-SCC changepoint density across the genome at four different bin sizes.** GenomeTrackSig was run genome-wise on 32 Lung-SCC samples with a bin size of either 100, 150, 200, or 300. Density plot across the genome of pooled changepoint positions in all samples is shown for each bin size analyzed.
(TIFF)

**S5 Fig. Number of recurrent changepoints identified at different sample thresholds.** Top: Number of recurrent changepoints identified across all cancer types depending on which number of samples is used as the threshold to determine which changepoints are considered 'recurrent.' Bottom: Number of recurrent changepoints identified in each cancer type compared to the sample threshold. Sample size and median tumor mutational burden is shown for each cancer type.
(TIFF)

# Acknowledgments

We thank Dr. Raluca Gordan and Harshit Sahay for helpful discussions on the chromatin accessibility analysis.

# Author Contributions

**Conceptualization:** Quaid Morris.

**Data curation:** Quaid Morris, Caitlin F. Harrigan.

**Formal analysis:** Caitlin Timmons, Caitlin F. Harrigan.

**Funding acquisition:** Quaid Morris.

**Resources:** Quaid Morris.

**Supervision:** Quaid Morris, Caitlin F. Harrigan.

**Writing – original draft:** Caitlin Timmons, Quaid Morris, Caitlin F. Harrigan.

**Writing – review & editing:** Caitlin Timmons, Quaid Morris, Caitlin F. Harrigan.

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
