## [Decision Letter · Decision Letter 0]

30 Sep 2022

Dear Ms. Timmons,

Thank you very much for submitting your manuscript "Regional mutational signature activities in cancer genomes" for consideration at PLOS Computational Biology. As with all papers reviewed by the journal, your manuscript was reviewed by members of the editorial board and by several independent reviewers. The reviewers appreciated the attention to an important topic. Based on the reviews, we are likely to accept this manuscript for publication, providing that you modify the manuscript according to the review recommendations.

We sincerely apologise for the delay in this review process. We have now finally received the comments from three reviewers and, based on their reccommendations, we would be glad to invite a minor revision of your manuscript to be considered for publication. The reviewers agree on the importance and novelty of the work, but they also believe that a few minor comments should be addressed in a final version of your manuscript before final acceptance. Particularly, based on reviewers' comments, we would like to ask the authors to address two of the main remaining comments raised by the reviewers:

1- Could the authors add a minimal set of basic simulations, possibly comparing with the previous method GenomeTrackSig, to show the accuracy of your method?

2- Could the authors revise the text according to reviewers' reccommendations, in particular adding a complete description of the method?

Sincerely,

Simone Zaccaria

Guest Editor

PLOS Computational Biology

Jason Papin

Editor-in-Chief

PLOS Computational Biology

We sincerely apologise for the delay in this review process. We have now finally received the comments from three reviewers and, based on their reccommendations, we would be glad to invite a minor revision of your manuscript to be considered for publication. The reviewers agree on the importance and novelty of the work, but they also believe that a few minor comments should be addressed in a final version of your manuscript before final acceptance. Particularly, based on reviewers' comments, we would like to ask the authors to address two of the main remaining comments raised by the reviewers:

1- Could the authors add a minimal set of basic simulations, possibly comparing with the previous method GenomeTrackSig, to show the accuracy of your method?

2- Could the authors revise the text according to reviewers' reccommendations, in particular adding a complete description of the method?

Reviewer's Responses to Questions

**Comments to the Authors:**

Reviewer #1: The authors have added additional details and analyses that make the results in this revised version of the manuscript even more interesting, and they have successfully addressed several reviewers’ comments. There are however some key comments that have not been appropriately addressed, especially related to the new algorithm, its benchmarking, and formal descriptions of the formal methods. Formal descriptions of methods are essential for both assessing and demonstrating the robustness of the proposed analysis and results, as well as for reproducibility. Therefore, these remaining comments should be appropriately addressed before the manuscript could be considered for final publication.

1. No simulation nor benchmarking analysis has been performed to assess and demonstrate the performance of GenomeTrackSig, as suggested by previous reviewers. While the authors highlight that the focus of the manuscript is on the outcome of the analysis, the results of the analysis cannot be believed if the accuracy and robustness of the proposed method are not clearly demonstrated. I would recommend the authors to consider using simulations to demonstrate the performance of the algorithm, as well as to address other reviewers’ comments, such as the impact of segment size, mutational burden, etc.

2. Related to the previous point and other reviewers’ comments, it would also be important to benchmark the results of GenomeTrackSig with those of previous methods. In particular, it would be important to demonstrate that the results obtained by GenomeTrackSig could have not been obtained in a straightforward way by simply using TrackSig (ie. without the new segmentation introduced by GenomeTrackSig). Moreover, while the authors have reported some differences with the previous method SigMa and have addeded a useful comparison section, these differences do not prevent the possibility to compare the basic performance of the two methods (GenomeTrackSig and SigMa) in estimating the spatial genomic distributions of the identified mutational signatures. Thus, comparing the performance of those methods using the proposed simulations would be important to demonstrate the impact of the new contribution in this manuscript.

3. The authors have introduced a new method section to better explain the new proposed algorithm. However, this section is still lacking formal mathematical descriptions; especially the entire section does not include a single equation to describe the proposed model and the relationship between the defined variables. For example, what is the probabilistic generative model on which the EM algorithm has been applied to (definition of random variables, related probabilities, etc.), or what is the relation linking the mutation matrix to the signature activities? Also, formal definitions are missing: what is the mathematical definition, domain and constraints of variables s_i? What is the definition of mutation type (I guess nucleotide context?)? What are the exact variables and equation used for applying the BIC criterion? What is the optimization function that defines an optimal segmentation? etc.

4. The current structure of the manuscript is confusing as the Result section describes the results of the analysis even before explaining or describing the new algorithm that have been developed to perform these analyses. Similarly to other manuscripts in this journal, it would be beneficial to start Results with a paragraph dedicated to providing a summary of the new proposed method.

5. Method section is substantially lacking several details and required descriptions, for example a full description of the processing pipelines applied to prepare the data and perform the analysis. Even more importantly, the formal descriptions of the sophisticated statistical analyses reported in the reviewers’ response (randomizations, generation of null distribution for change points, bootstrapping, etc.) are not reported in the Method section. I would invite the authors to carefully add in this section the formal details and descriptions of all the performed analyses, as well as the data processing and the new proposed algorithm.

Reviewer #2: I was a reviewer for RECOMB-CCB. The authors have properly addressed all reviewers’ comments, providing additional experiments and evidence supporting their main conclusions. The new version of the manuscript greatly improves over the previous version.

I have only a minor comment. The "Related work" section describes is 1 paragraph describing the related method SigMa, and coming after the Results section it feels a bit out of place (since the method is not described as a result). I think it makes more sense to incorporate such paragraph in the Introduction or in the Materials and Methods section.

Reviewer #3: Review of "Regional mutational signature activities in cancer genomes" submitted to PLOS Computational Biology

September, 2022

One of my students has previously helped with reviewing this paper for conference and she informed me that all of her comments are now addressed. The paper was presented at RECOMB-CCB earlier this year in San Diego and we attended the interesting talk given by the first author. Considering that this is a journal version, I would kindly ask the authors to do the following:

1. The paper would benefit from a substantial revision and extension of the presentation. This primarily refers to extending Introduction and Methods sections to make them more accessible to the broader audience. Dependence on TrackSig and other previous works should be minimal. Some descriptions can be moved to Supplementary if needed. I totally understand that this was difficult to do previously for the conference submission due to page limits, deadlines etc. and this was to some extent acceptable due to very specialized audience at the conference, but I am concerned that understanding the paper might be challenging for less specialized readers, which could limit its potential impact.

2. The documentation at the github repository (https://github.com/morrislab/GenomeTrackSig) is currently incomplete. To give a simple example, no explanation of the input files is provided. Please provide more detailed and complete documentation for everything. Also, I recommend you to provide (in README) details of who should be preferably contacted and at what email address in case that a potential user has some questions, but I leave this as optional considering that some contact details will be provided in the main paper.

3. In addition to the previous point, I noticed that there is file Example_cna.txt in extdata folder (https://github.com/morrislab/GenomeTrackSig/blob/master/inst/extdata/Example_cna.txt). While it is not clear from the documentation what purpose this file serves and whether it is needed at all (I do not see it being used in any of the example commands shown in README so it is possible that this is something that the authors forgot to delete from the repository), its last column seems to be consisting only of 2's. If this is true and the file is a part of the input/output, then this suggests that the provided examples can perhaps be improved to cover a wider range of permissible inputs. Please double check this.

**Have the authors made all data and (if applicable) computational code underlying the findings in their manuscript fully available?**

Reviewer #1: Yes

Reviewer #2: Yes

Reviewer #3: **No: **It is unclear how the codes should be used, but this is something that can be fixed.

PLOS authors have the option to publish the peer review history of their article (what does this mean?). If published, this will include your full peer review and any attached files.

Reviewer #1: No

Reviewer #2: No

Reviewer #3: No

Figure Files:

Data Requirements:

Reproducibility:

References:

---

## [Editor Report · Decision Letter 1]

14 Nov 2022

Dear Ms. Timmons,

We are pleased to inform you that your manuscript 'Regional mutational signature activities in cancer genomes' has been provisionally accepted for publication in PLOS Computational Biology.

Best regards,

Simone Zaccaria

Guest Editor

PLOS Computational Biology

Jason Papin

Editor-in-Chief

PLOS Computational Biology

---

## [Editor Report · Acceptance letter]

26 Nov 2022

PCOMPBIOL-D-22-01048R1 

Regional mutational signature activities in cancer genomes

Dear Dr Morris,

I am pleased to inform you that your manuscript has been formally accepted for publication in PLOS Computational Biology. Your manuscript is now with our production department and you will be notified of the publication date in due course.

With kind regards,

Zsofia Freund
